# Fragmentation of production amplifies systemic risks from extreme events in supply-chain networks

Célian Colon[1]*, Åke Brännström[1,2], Elena Rovenskaya[3,4], Ulf Dieckmann[1]

**1** Evolution and Ecology Program, International Institute for Applied System Analysis, Laxenburg, Austria, **2** Department of Mathematics and Mathematical Statistics, Umeå University, Umeå, Sweden, **3** Advanced Systems Analysis Program, International Institute for Applied System Analysis, Laxenburg, Austria, **4** Optimal Control Department, Faculty of Computational Mathematics and Cybernetics, Moscow State University, Moscow, Russia

* celian.colon@polytechnique.edu

**Data Availability Statement:** All relevant data are within the paper and its Supporting Information files.

## Abstract

Climatic and other extreme events threaten the globalized economy, which relies on increasingly complex and specialized supply-chain networks. Disasters generate (i) direct economic losses due to reduced production in the locations where they occur, and (ii) to indirect losses from the supply shortages and demand changes that cascade along the supply chains. Firms can use inventories to reduce their risk of shortages. Since firms are interconnected through the supply chain, the level of inventory hold by one firm influences the risk of shortages of the others. Such interdependencies lead to systemic risks in supply chain networks. We introduce a stylized model of complex supply-chain networks in which firms adjust their inventory to maximize profit. We analyze the resulting risks and inventory patterns using evolutionary game theory. We report the following findings. Inventories significantly reduce disruption cascades and indirect losses at the expense of a moderate increase in direct losses. The more fragmented a supply chain is, the less beneficial it is for individual firms to maintain inventories, resulting in higher systemic risks. One way to mitigate such systemic risks is to prescribe inventory sizes to individual firms—a measure that could, for instance, be fostered by insurers. We found that prescribing firm-specific inventory sizes based on their position in the supply chain mitigates systemic risk more effectively than setting the same inventory requirements for all firms.

## Introduction

Supply disruptions raise significant concerns for businesses [1–6], and their adverse financial effects have been empirically established [7–9]. Trigger events include industrial accidents, natural disasters, and climatic extreme events, whose currently changing patterns are posing a major threat [10]. Localized disruptions may cascade from one firm to another and cause significant economic losses in distant locations [11, 12]. For example, such ripple effects gave the 2011 earthquake in Tokohu, Japan, and the 2011 floods in Thailand a global reach [13–15].

**Funding:** CC received a stipend from the International Institute for Applied System Analysis (IIASA, http://www.iiasa.ac.at/), through the Young Scientists Summer Program (YSSP). The funders had no role in study design, data collection and analysis, decision to publish, or preparation of the manuscript.

**Competing interests:** The authors have declared that no competing interests exist.

Even minor accidents may generate sizeable impacts on supply chains [e.g., 12], especially when they affect the production of very specific inputs [9].

Systemic risk characterizes losses resulting from such cascading disruptions. This kind of risk is more commonly associated with financial systems [16], but generally emerges in networked systems in which the state of one node depends on the activities of the others. Production systems have indeed coalesced into a global network as a result of the large-scale transformations that have developed over the past decades, namely, offshoring, outsourcing, and vertical specialization [17–19]. In many sectors, production has been split into multiple sequential stages operated by geographically diverse firms. Many manufactured goods are now assembled from a large number of outsourced components. These trends have resulted in longer and more interconnected supply chains, and firms often report a very limited visibility of the supply chains of which they are a part [20–22]. Some features of supply chains can be tracked with sectorial input–output tables [e.g., 23] or supplier–buyer data [e.g., 24]. The consequences for systemic risk of the greater structural complexity of production networks have started to be investigated empirically [15] and theoretically [25–27].

Another effect of outsourcing and vertical specialization is the increased fragmentation of supply chains, whereby production stages become highly segmented and each segment is run by a legally distinct firm [17]. Fragmentation creates additional challenges to the mitigation of systemic risk. Even though firms are implementing risk-mitigating measures at their level—for example, through inventories, supplier diversification, or operational flexibility—their exposure to systemic risk also depends on the measures implemented by others. For instance, in a lean management perspective, manufacturers reduce their inventories by working with a few highly reliable suppliers. Conversely, if a supplier decides to engage in more risk-prone operations, its clients may, in response, build larger inventories or find alternative suppliers. In a supply chain, even if firms aim to design a risk-mitigating strategy that improves their own profitability, in practice this strategy depends on the strategies implemented by others. If then a supply chain becomes more fragmented, its exposure to systemic risk depends on an even larger number of other firms. How does this proliferation of distinct yet interdependent risk-mitigating strategies affect the overall level of systemic risk in a supply chain, as well as its resilience?

The tools and methods of game theory have traditionally been applied to the study of such strategic interactions between profit-maximizing agents. This analytical framework has also been used in operations research and management science to determine the optimal strategies for firms to mitigate supply disruptions [28] and to identify ways of fostering cooperation along supply chains [29, 30]. The assumption of strong rationality, according to which agents instantaneously process the full decision trees of all agents over an infinite time horizon, has, however, limited the application of game theory to small or idealized supply chains, in which the link between systemic risk and fragmentation cannot be addressed.

To overcome this limitation, here we develop a model of supply chains based on evolutionary dynamics of social learning occurring in games unfolding on networks [31, 32]. Firms follow simple behavioral rules to explore and adjust risk-mitigating strategies in order to increase their profits. We examine the strategic interactions in complex supply chains and study the effects of fragmentation. In our model, final producers attempt to fulfill a fixed demand from households. They produce goods using inputs from suppliers, who themselves purchase inputs from other firms, thus forming supply chains. The production of every single firm is subject to randomly occurring failures or disasters. Firms use inventories to mitigate the risk of supply disruptions. According to the level of systemic risk they experience, they adjust the rate at which they order goods from their suppliers so as to increase their expected profits. Where supplies are ordered beyond equilibrium production needs, we speak of overordering.

Through this stylized, 'toy' model, we qualitatively characterize the impact of supply-chain fragmentation on risk reduction. Our theoretical analysis includes two special cases: in fully fragmented supply chains, each firm aims to increase its own profit, whereas in fully integrated supply chains, all firms aim to increase the total profit of the supply chain. Between these extremes, we study the full range of intermediate fragmentation scenarios.

## Model

### Supply chains as input–output networks

We model a supply chain as a directed acyclic network of $n$ firms with adjacency matrix $M = (m_{ij})$, such that $m_{ij} = 1$ if firm $i$ is supplying goods to firm $j$ and $m_{ij} = 0$ otherwise. No firm supplies to itself; hence, $m_{ii} = 0$ for all $i$. Firms that have no incoming links in the network produce using raw materials and are called primary producers; at the other end of the chain, firms that do not have outgoing links to other firms sell their production to households and are called final producers. Since the supply chain is acyclic, all other firms are intermediary producers. The economic activity is driven by a fixed demand from households, represented by a vector $D = (d_1, d_2, . . ., d_n)$, such that $d_i = 1/n_f$ if firm $i$ is one of the $n_f$ final producers and $d_i = 0$ otherwise. To meet this demand, final producers order inputs from their suppliers, which themselves order inputs from their suppliers, and so on, down to the primary producers. Each firm has a linear production function, that is, it transforms a quantity $x$ of input into a quantity $zx$ of output where parameter $z > 1$ is called productivity. For simplicity, we assume that all firms have the same productivity. All quantities are expressed in monetary terms. Inputs are assumed to be fully substitutable and storable with durability $v$. The latter means that, at the end of each time step, while all inputs that a firm did not use are added to its inventory, a portion $1-v$ of this inventory becomes obsolete and is discarded. In the absence of external perturbations, firms order the exact quantity of inputs required to meet the demand they face. We consider the input–output matrix $A = (a_{ij})$, where $a_{ij}$ describes how much input from firm $j$ is needed by firm $i$ to produce one unit of output. Firms equally divide the total amount of inputs they need between their suppliers, such that $a_{ij} = m_{ij}/(z\, s_i)$, where $s_i$ is the number of suppliers of firm $i$. Using linear algebra, we derive that the vector of production targets $Y = (y_1, y_2, . . ., y_n)$, which is equal to $(I-A)^{-1}D$, where $I$ is the identity matrix. In the absence of supply disruptions, each firm produces its production target $y_i$. Under these conditions, the profit of each firm, $\pi_{0i}$, defined as sales minus inputs costs, equals $y_i - y_i/(z\, s_i)$; other costs involved in production, such as labor and capital costs, are considered to be fixed and are thus not needed for our model.

### Dynamic responses to external perturbations

To model supply disruptions, we assume that in each time step, each firm loses its entire production with probability $p \in [0,1]$, called the failure rate, which is the same for all firms. Firms that experience such a perturbation are unable to supply goods to their clients, who may as a result lack the necessary inputs to meet their demands. Rationing may occur and the disruption may cascade further along the supply chain, leading to profit losses by firms located downstream from the initial failure. To mitigate this risk, firms build inventories of inputs, denoted by $h_{t,i}$, by ordering at each time step some extra units from their suppliers. Specifically, if firm $i$ faces demand $y_i$, it constantly orders a quantity $y_i(1+r_i)/(zs_i)$ of inputs from each of its suppliers, in which $r_i > 0$ is the overordering rate specific to each firm and constant over time. Overordering raises input costs, such that when no perturbation occurs, profit is reduced; but in the event of a supply disruption, inventoried inputs compensate for potential losses. With overordering, the input–output coefficients become $a_{ij} = m_{ij}(1+r_i)/(zs_i)$; they are used to

compute the production target $y_i$ of each firm. Because, over time, production disruptions occur randomly, the inputs received by firms, denoted by $x_{t,i}$, can be smaller than the ordered quantities, inventories $h_{t,i}$ varies over time, and the actual production levels of firms, denoted by $y_{t,i}$, can be smaller than the targets. All production is always sold, such that profits $\pi_{t,i}$, which is defined by sales minus costs, is equal to $y_{t,i}-x_{t,i}$. To evaluate the impact of firms' over-ordering on their profits, we conduct numerical simulations and calculate the average profits of each firm over a long time-horizon $T$, defined by $\bar{\pi}_i = \langle\pi_{t,i}\rangle_{0 \leq t \leq T}$. For simple supply chains, we derive reduced-form dynamical equations; see *S1 Section* in *S1 File*. In addition, for specific classes of layered supply chains, we have developed an algorithm that enables us to calculate the exact expected value of firms' profits; see *S5 Section* in *S1 File*.

The aggregate losses of the entire supply chain measure the decrease in all firms' profit, using as a baseline the case without perturbations and without overordering,

$$L = \sum_i (\pi_{0i} - \bar{\pi}_i).$$

Aggregate losses can be split into direct losses $L_D$, incurred by perturbed firms, and indirect losses $L_I$, resulting from the propagation of disruptions throughout a supply chain. In a simulation, direct losses can be measured by summing the profit losses of firms only when they are externally perturbed. Indirect losses are then obtained as the difference between aggregate losses and direct losses.

To evaluate the mitigation success $S$ produced by a vector of overordering rates $R = (r_1, r_2, \ldots, r_n)$, we measure the relative change in the indirect losses compared to a counterfactual case in which firms ignore disruption risks and do not overorder, that is, $R_0 = (0, 0, \ldots, 0)$,

$$S(R) = \frac{L_I(R) - L_I(R_0)}{L_I(R_0)}.$$

## Adjustment of overordering and supply-chain fragmentation

Firms can adjust their overordering rates to achieve a certain objective. In our analyses, we assume firms to be risk neutral. We consider a scenario of full fragmentation, in which firms overorder at a rate that maximizes their own expected profits. Since the overordering rate chosen by one firm typically affects the profits of other firms, firms interact strategically, which means that each firm tries to respond optimally to the decisions of the others. The vector of overordering rates at which no firm can unilaterally increase its profit corresponds to a Nash equilibrium, which we denote by $R^*$. To study this situation, we follow the dynamics of strategy evolution based on a stochastic gradient-ascent algorithm, whereby each firm iteratively tries and tests overordering rates and adopts the ones that increase its profit. The details are presented in *S6 Section* in *S1 File*. We analytically demonstrate the existence and uniqueness of this Nash equilibrium for simple supply chains in *S4 Section* in *S1 File*. Numerical evidence is provided for the general case; see *S7 Section* in *S1 File*. We then evaluate the mitigation success $S^*$ that is achieved when all firms adopt the Nash-equilibrium overordering rates.

Next, we contrast the full-fragmentation scenario with scenarios of partial fragmentation. Specifically, we partition the set of firms into groups, such that, within a group, each firm adjusts its overordering rate to maximize the group's profit. A group can be composed only of adjacent firms, i.e., each firm in a group has to be the supplier or the client of another firm in the group. A supply chain with $g$ groups has a fragmentation of $f = (g-1)/(n-1)$. We generate a large number of random group configurations and study how the fragmentation $f$ affects the mitigation success $S^*$.

## Numerical and analytical investigations

General results are derived from numerical investigations of large ensembles of random supply chains generated using the classic algorithm for Erdős–Rényi graphs restricted to the upper triangle of the adjacency matrix [33, 34]. As a baseline, we have chosen supply chains with $n = 30$ firms and an average of $c = 2$ suppliers per firm, performing robustness checks for $10 \leq n \leq 100$ and $1 \leq c \leq 4$. In large scale datasets on supplier–buyer relationships, which concerns complex economies, namely the USA and Japan, the number of crucial suppliers per firm is found between 3 and 4 [9, 24, 35–37]. Our choice of network topology leads to supply chains whose lengths—defined by the average trophic level of the final producers—range between 3 and 6, which is reasonable compared to real supply chains.

Because we are interested in the general behavior of the model, we explore the firm-parameter space $1 \leq z \leq 5$, $0 \leq v \leq 100\%$, and $0 \leq p \leq 100\%$, as well as cases with distributed values of the firm-level parameters, in which a different combination of $z$, $v$, and $p$ is given to each firm. We also explore the full fragmentation range of $0 \leq f \leq 100\%$. To illustrate the results, we selected representative examples by using the following benchmark values: $z = 2$, $v = 50\%$, and $p = 10\%$. A productivity value of 2 is consistent with the average output over input ratio found in national input–output tables; e.g., a value of 2.2 is found for the 2015 table for the USA provided by the OECD [38]. Durability widely varies according to the type of products, hence our choice of 50%. Last, a 10% failure rate implies that firms are on average unable to produce 5.2 weeks per year, due for instance to disasters, strikes, epidemics, or accidents.

To advance the understanding of the role of each parameter in the overordering decisions, we perform analytical studies of simpler supply chains: a single producer supplied by multiple firms and a class of multi-layered networks. These analytical results, given in S1 to S5 Section in S1 File, are qualitatively similar to the results in the general case and help facilitate its interpretation. Table 1 summarizes the mathematical symbols used in this section. The code of the model is available online at https://github.com/ccolon/supply-chain-fragmentation.

## Results

### Optimal inventory sizes increase along supply chains and are highly context dependent

Through strategy evolution, firms adapt their overordering rates to increase their profits; as an example, see the colored curves in Fig 1A for the fully fragmented supply chain in Fig 1B. In general, the speed and outcome of strategy evolution differ among firms. For some firms, holding inventories only weakly improves their profits, so that they slowly and gradually adjust their overordering rates. The profit of other firms, in contrast, is highly sensitive to their overordering rates, which thus rapidly increase from their initially low values; see the first time-steps in Fig 1A. These rates are then readjusted to account for the effects of the inventories adopted by the other firms. As in Fig 1A, strategies may largely fluctuate around stationary values. This is because supply-chain perturbations are stochastic, so the profit associated with a given overordering rate varies over time. The impact of this stochasticity on strategy evolution is discussed in S6 Setion in S1 File.

Fig 1b shows how the resulting overordering rates are distributed across the supply chain. In this instance, more than a half of the firms do not overorder at all, mainly those located upstream that have five or fewer total suppliers—total here means that suppliers of all tiers are counted. This observation is in accordance with the distributions of overordering rates obtained from an ensemble of 2,000 fully fragmented random supply chains with the same number of firms and connectivity—defined as the average number of suppliers per firm—, but

**Table 1. Mathematical symbols used in the model section and their definitions.**

| Symbol | Definition |
|---|---|
| *Supply-chain parameters* | |
| $n$ | Number of firms in the network |
| $c$ | Average number of suppliers per firm |
| $M = (m_{ij})$ | Adjacency matrix |
| $s_i$ | Number of suppliers of firm $i$ |
| $g$ | Number of groups in the supply chains. Within a group, each firm adjusts its overordering rate to maximize the group's profit |
| $f$ | Supply chain fragmentation, i.e., $f = (g-1)/(n-1)$ |
| *Free firm-level parameters* | |
| $z$ | Productivity, i.e., how much monetary unit of output can be produced from one monetary unit of input |
| $v$ | Durability of the inventories, i.e., a fraction $1-v$ becomes obsolete at each time step |
| $p$ | Failure rate, i.e., probability that a firm get perturbed—i.e., become unproductive—during at each time step |
| *Evolutionary variables* | |
| $R = (r_1, r_2, \ldots, r_n)$ | Overordering rates |
| $R^*$ | Overordering rate at the evolutionary equilibrium |
| *Input-output equilibrium* | |
| $D = (d_1, d_2, \ldots, d_n)$ | Fixed final demand |
| $A = (a_{ij})$ | Input–output matrix, $a_{ij} = m_{ij}(1+r_i)/(z\,s_i)$ |
| $Y = (y_1, y_2, \ldots, y_n)$ | Production targets, $Y = (I-A)^{-1}D$ |
| $y_i(1+r_i)/(z\,s_i)$ | Quantity ordered by firm $i$ to its suppliers |
| *Time-dependent variables* | |
| $x_{t,i}$ | Input received by firm $i$ at time $t$ |
| $y_{t,i}$ | Output produced and sold by firm $i$ at time $t$ |
| $h_{t,i}$ | Inventory of firm $i$ at time $t$ |
| *Observables* | |
| $\pi_{t,i}$ | Profit of firm $i$ at time $t$, i.e., sales minus inputs |
| $\pi_{0,i}$ | Profit of firm $i$ at time $t = 0$, at which no perturbation is applied |
| $\bar{\pi}_i$ | Average profit of firm $i$ over a time horizon $T$ |
| $L$ | Aggregate loss, i.e., $\sum_i (\pi_{0i} - \bar{\pi}_i)$ |
| $L_D$ | Direct loss, i.e., loss of profit of firms directly affected by a perturbation |
| $L_I$ | Indirect loss, i.e., loss of profit of firms not directly affected by a perturbation |
| $S(R)$ | Mitigation success associated with the overordering rates $R$, i.e., $S(R) = [L_1(R)-L_1(R_0)]/L_1(R_0)$, where $R_0 = (0, 0, \ldots, 0)$. |
| $S^*$ | Mitigation sucess at the evolutionary equilibrium, $S^* = S(R^*)$ |

different connection structures; see Fig 1c. Given that the inflow of raw materials is not subject to perturbations, primary producers never overorder. As we move downstream, firms become increasingly subject to supply disruptions because each of their suppliers, direct and indirect, is a potential source of disruption. Risks gradually accumulate until thresholds are crossed and overordering becomes beneficial. Using simpler supply chains, we demonstrate the existence of such thresholds analytically and show how they arise from the limited number of independent suppliers and from the non-durability of goods; see *S2-S4 Figs in S1 File*. We find that the threshold value of total suppliers—five in Fig 1c—diminishes with higher failure rates, more productive firms, and more durable goods.

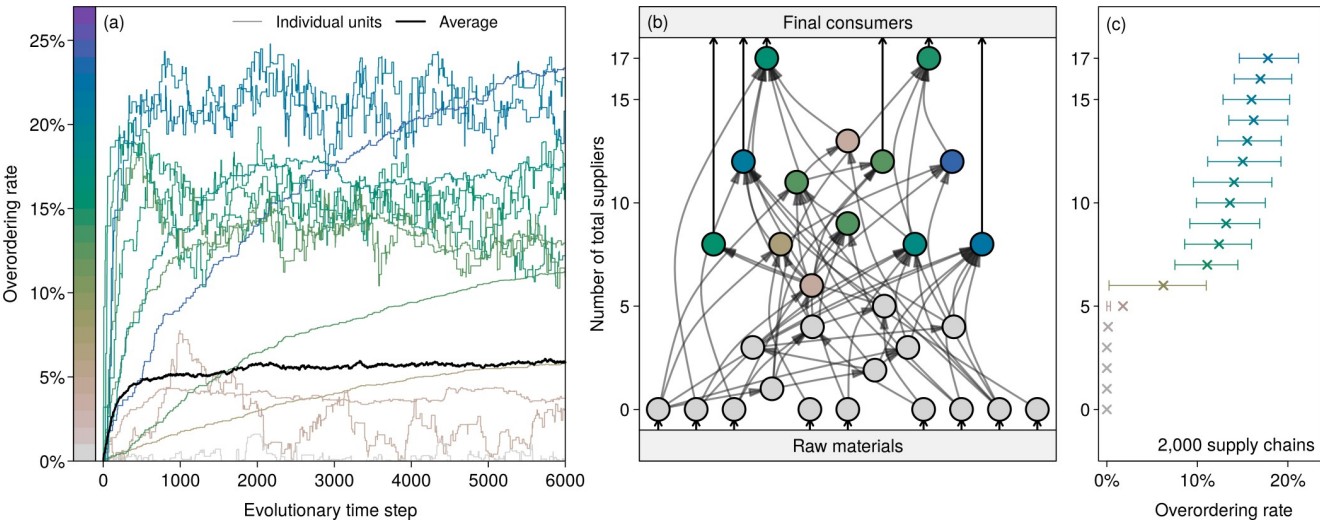

**Fig 1. Heterogeneous overordering patterns emerge from strategic interactions among the firms in a supply chain.** Panel (a) presents the evolution of the overordering rates of the 29 firms of the supply chain displayed in panel (b). Firms start without overordering, then adjust their rates to increase their profits. The thick black curve shows the average overordering rate. Each of the other curves corresponds to a firm; they are colored according to the final overordering rate adopted by the firm, as shown by the color bar on the left. In panel (b), firms are nodes, colored to correspond to the curves of panel (a). The 61 grey arrows represent supplier–buyer interactions, while the black arrows indicate the flows that go in and out of the chain: inflows of raw materials and outflows of final goods. The vertical positions of the firms are proportional to their number of total suppliers. Panel (c) uses the same vertical axis to compare the results for this specific supply chain with statistics from an ensemble of 2,000 random supply chains with the same number of firms and connectivity. Crosses indicate means and whiskered bars indicate interquartile intervals.

As seen in Fig 1c, beyond this threshold, there is a positive relationship between overordering rates and the number of total suppliers, which shows that, on average, having an additional supplier exposes a firm to a larger risk. However, the horizontal dispersion of the results implies that this simple rule does not apply to all situations. A supplier that maintains large inventories abates the risk of supply disruption for its clients, who may then overorder less. In Fig 1b, for instance, the two final producers, shown in green, each with 17 total suppliers, overorder less than their suppliers, shown in blue, each with just 12 total suppliers. This type of interaction also explains why strategy evolution in Fig 1a may reverse direction. In specific supply chains, this mechanism may lead to surprising patterns, as shown in *S6 Fig* in *S1 File*. Additional results on correlations between the position of firms and their optimal overordering rates are presented in *S10 Fig* in *S1 File*.

## Inventories greatly alleviate disruption cascades and moderately increase direct losses

Without inventories, most economic losses provoked by external perturbations are indirect. All supply disruptions propagate downstream, leading to large disruption cascades. Systemic risk is higher in longer and more interconnected supply chains. For instance, in the complex supply chain in Fig 1B, a random external perturbation disrupting one firm generates profit losses for, on average, 5.4 other firms, with indirect losses being more than twice as large as direct ones; see Fig 2A. Note that because we assume inputs to be perfectly substitutable, the indirect losses will be even larger in situations with only partial substitutability. Once firms have adopted the overordering rates that maximize their profits, they maintain inventories, which act as buffers against disruption cascades; in Fig 2A, for instance, the average size of disruption cascades is more than halved, down to 2.1 other firms. In particular, the chance that a final producer experiences an input shortage following the disruption of a primary producer is

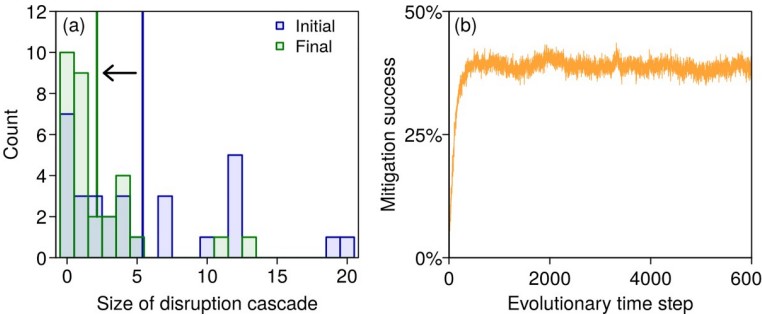

**Fig 2. Overordering diminishes disruption cascades and mitigates risks.** Both panels refer to the supply chain in Fig 1a and 1b. Panel (a) shows how the distribution of the size of disruption cascades changes between the initial state, shown in blue, in which no firms overorder, and the outcome of strategy evolution, shown in green. The thick vertical lines indicate the means, moving from 5.4 down to 2.1 firms. The size of a disruption cascade is the number of firms affected by supply shortages following an external perturbation; the distribution is obtained by perturbing each firm one by one. Panel (b) displays how mitigation success, which measures the relative reduction in indirect losses, changes during the strategy evolution in Fig 1a.

reduced from 100% to 55% for the supply chain in Fig 1b, leading to a 39% drop in indirect losses; see Fig 2B. We generally observe an increase in direct losses associated with overordering. Because they buy more inputs, the overordering firms are running costlier production processes. When externally perturbed, they lose their production but still have to pay for the extra inputs they ordered, thus experiencing greater direct losses. This negative impact remains moderate compared with the mitigation of indirect losses. Even in a fully fragmented supply chain, decentralized profit-driven decisions thus allow inventories to be built that decrease systemic risk.

## Firms hold less inventories in more fragmented supply chains and generate systemic risks

When firms are grouped together and aim at increasing their group's profit instead of their own, they overorder more. This behavior induces larger inventories and leads to stronger risk mitigation. This result is valid over the whole parameter space of our model and is robust to changes in the supply-chain structure. It is shown in Fig 3 for six classes of networks. The mitigation success, which measures the relative reduction in indirect losses, is the smallest for the case of full fragmentation and the largest for the case of full integration. For supply chains with 30 firms and two suppliers per firm, for instance, full integration helps increase the mitigation success by two-thirds compared to full fragmentation. This stronger mitigation comes from the internalization of positive externalities. When a firm overorders, it generates positive outcomes both upstream and downstream: upstream firms receive a larger demand, while downstream firms are less exposed to disruption cascades. These benefits do not cost them anything: they are externalities. If a firm is grouped together with some of its suppliers or customers, a share of these indirect benefits feeds back into its decision-making process, rendering overordering more attractive. Integration therefore enables the internalization of the positive externalities associated with overordering, resulting in larger inventories and lower risks of supply disruptions. The steepest slopes we find in Fig 3 for low levels of fragmentation show that, in otherwise fully integrated supply chains, a slight degree of fragmentation has a high impact. The vertical ordering of the curves in Fig 3 also suggests that mitigation is more effective in more interconnected supply chains. Although such systems are prone to larger disruption cascades, firms are better able to manage them because they have access to a larger

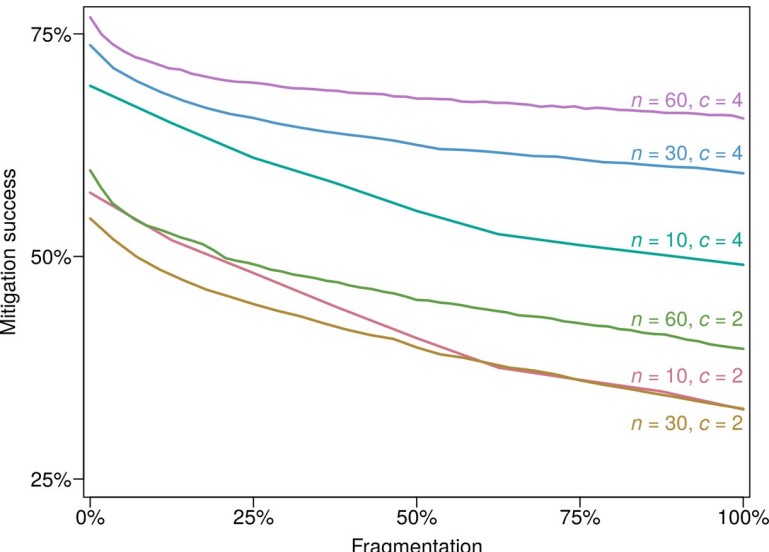

**Fig 3. Supply-chain fragmentation disincentivizes inventories and reduces risk mitigation.** The figure shows how mitigation success changes with fragmentation for six classes of supply chains, defined by the number of firms, $n$, and the average number of suppliers per firm, $c$. Fragmentation is defined as $(g-1)/(n-1)$, where $g$ is the number of groups of integrated firms. Each curve shows the average over 20 random supply chains, and for each of them, $10n$ group configurations are assessed. The dispersion of the results is shown in *S7 Fig in S1 File*.

number of fully substitutable suppliers. The positive impact of supplier diversification is shown in the analytical results of *S2-S4 Figs in S1 File*.

## Systemic risks are best mitigated in integrated supply chains with durable inventories

While systemic risk is robustly reduced by supply-chain integration, the exact level of optimal mitigation depends on three firm-level characteristics: productivity $z$, failure rate $p$, and durability $v$. The dependence of the mitigation success on these parameters exhibits some regularities; these are shown in Fig 4 for fully integrated supply chains. First, when the failure rate $p$ is high compared to the productivity $z$, supply chains can become unproductive. This occurs when $p \geq 1-1/z$, independent of the durability $v$ and the supply-chain structure; see the grey regions in the three panels of Fig 4. There, firms use on average more than one unit of input to produce one unit of output, i.e., they are unproductive. Inside the productive regions, the

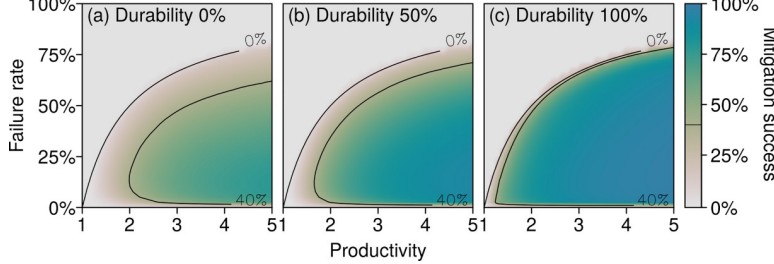

**Fig 4. Durable goods facilitate robust risk mitigation.** The three panels present the mitigation success of integrated supply chains according to productivity $z$, failure rate $p$, and for three levels of durability: (a) 0%, (b) 50%, and (c) 100%. The 0% contour is the boundary between productive and unproductive situations. The results are averaged over 10 random supply chains with 30 firms and an average of two suppliers per firm.

mitigation success peaks for intermediate failure rates; see the changes along vertical transects in Fig 4. When firms become more productive—moving rightward within each panel—or when goods become more durable—moving rightward across the panels—this peak gradually flattens out and becomes a plateau; see in particular Fig 4C for $z \geq 2$. With durable goods and in the productive region, keeping an inventory is always profitable. Any inventory will eventually be used during a disruption to avoid losing sales, with no extra costs. In that case, the mitigation success is high for any failure rate. With less durable goods, the cost of maintaining an inventory is no longer proportionate to sales. For high failure rates, very large overordering rates are needed. Building such large inventories may become unprofitable, so that the mitigation success drops; see the upper limit of the productive regions in Fig 4A and 4B. On the other hand, for low failure rates, firms may prefer to undergo occasional disruptions rather than permanently maintain inventories; hence, we observe weaker mitigation successes at the bottom of Fig 4A–4C. The mechanisms underpinning this behavior can be analytically assessed in simple supply chains; see *S2 and S3 Figs in S1 File*.

## Inventories calibrated on network analyses help reduce systemic risks

So far, we have elicited the optimal overordering rates through a decentralized process that takes into account strategic interactions between firms. We now examine whether a simple mapping of the supply chain can help decision-makers allocate, in a top-down manner, some reasonable overordering rates, which could be imposed through regulations or promoted through positive or negative incentives. To explore such options, we investigate the differential suitability of ten different indicators that capture information on the position of each firm in the supply chain; see *S2 Table in S1 File* for a full list and definitions. Each indicator—for instance, the number of suppliers—describes a firm $i$ by a value $s_i$. To compare across indicators, we center and normalize the values of $s_i$ for each indicator, so that the resulting values $s_i'$ average to 0 and have a standard deviation of 1. We then determine the overordering rate of each firm $i$ as $r_i = a + bs_i'$, where $a$ is the prescribed average overordering rate and $b$ is an elasticity factor scaling the indicator's effect. For each indicator, we pick the values of $a$ and $b$ that lead to the highest mitigation success. Fig 5A shows the results for the three most successful and the three least successful indicators, and Fig 5B compares the maximum mitigation successes obtained over a large ensemble of random supply chains. While these quantitative results depend on the choice of parameters, they characterize a qualitative behavior that is general for our model.

The three least-performing indicators fail to significantly outperform a uniform allocation, whereby all firms implement the same overordering rate. These are the number of clients and two centrality measures from network theory: closeness and betweenness centrality [39]. In contrast, three other indicators significantly improve the mitigation outcome: the number of direct suppliers, the number of total suppliers, and the supply-chain level. The latter is formally equivalent to the trophic level used in ecology [40]: it is the average number of links connecting a firm to primary producers, taking into account all possible pathways. This measure has, to date, not been used to describe the position of firms in supply chains. In our model, it captures the most appropriate information to allocate the mitigation efforts. For the specific, yet representative, case displayed in Fig 5A, the average overordering rate at which the mitigation success peaks is similar across the indicators. This similarity illustrates that the stronger mitigation success of the top three indicators is not merely due to larger risk-mitigating efforts, but to more appropriately allocated ones. In addition, the top three indicators significantly reduce systemic risk compared to decentralized decisions in fragmented supply chains; see the right-hand edge of the sloping histogram bar of Fig 5B. Nevertheless, even the supply-chain level cannot reach the mitigation success achieved when supply chains are fully integrated; see the

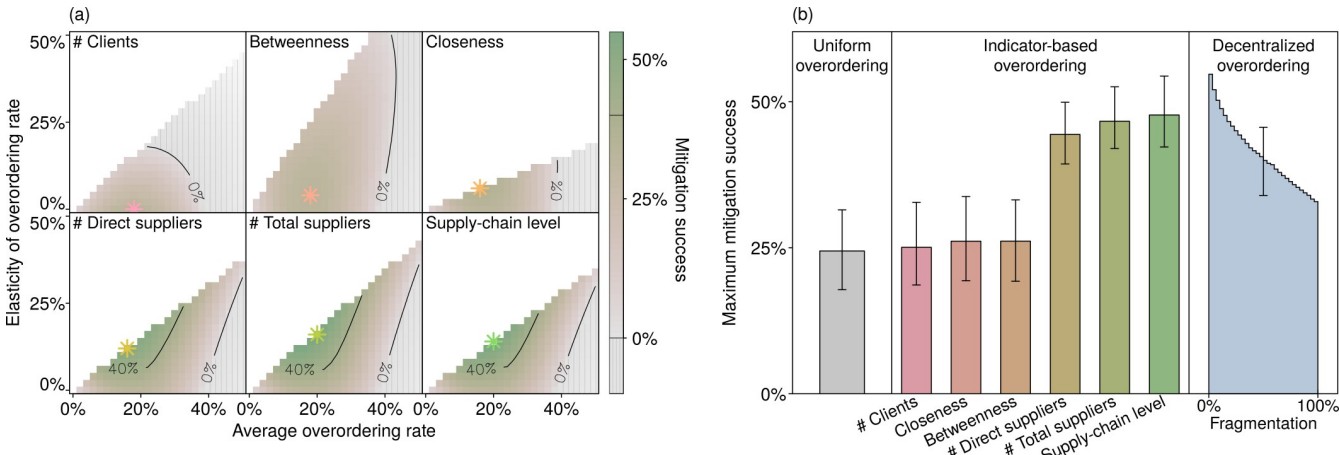

**Fig 5. Network indicators help allocate inventories.** Panel (a) shows, for a specific supply chain with $n = 30$ firms and an average of two suppliers per firm, the mitigation success of six indicators. In each subpanel, the $n$ values of the indicator are centered on 0, normalized to a standard deviation of 1, and then mapped into $26^2$ vectors of overordering rates, with the horizontal axis indicating the average rate and the vertical axis the elasticity, each being stepped in 26 levels. For high elasticities, some rates may fall below 0%, implying that some firms should underorder. Such combinations are removed from further study, as indicated by the upper-left white regions. The star symbols pinpoint the maximum mitigation successes. Panel (b) displays the maximum mitigation successes of the six indicators averaged over 200 random supply chains. The whiskered bars indicate the interquartile ranges. These mitigation successes (middle) are contrasted with the uniform allocation of overordering rates (left), whereby all firms implement the same rate, and with the outcomes of decentralized strategy evolution (right), whereby all firms maximize their own profits.

left-hand edge of the same bar. For the chosen parameter values, the average maximum mitigation success reached by the supply-chain level is 48%, achieved with an average overordering rate of 18%, whereas full integration reaches 56% mitigation success with an average overordering rate of 14%. This comparison demonstrates that the cooperative and decentralized optimization process that takes place in the full integration scenario mitigates risk not only more effectively—delivering a greater mitigation success—but also more efficiently—keeping smaller inventories. In addition, the peaks of Fig 5A are relatively narrow, suggesting that wrong decisions on the overall overordering efforts, or inaccurate information on a firm's position, are causing rapid deteriorations in mitigation success. Combining indicators does not significantly improve the mitigation success. A more detailed statistical analysis on the information provided by the ten indicators is presented in *S10 Section* in *S1 File*.

## Discussion

Systemic risk occurs due to cascading failures that propagate far through economic systems, resulting in potentially large financial losses. Here we have demonstrated for the first time that, under very general conditions, supply-chain fragmentation exacerbates systemic risk. The more fragmented a supply chain is, the more likely small disruptions are to propagate and become amplified. This additional risk arises from the strategic interactions among individual firms. In fragmented supply chains, the system-level benefits of inventories become externalities in the economic sense, so that individual firms have less incentive to maintain them. This theoretical finding is consistent with ample empirical observations that inventories have been shrinking during the outsourcing boom of recent decades. Although these practices have allowed firms to be more competitive, we argue that—if not carefully monitored, measured, and managed—they are bound to aggravate systemic risks in supply chains.

In the context of climate change, with shifting patterns of weather-related extreme events [10], supply chains are likely to face increasing levels of perturbations. Our findings suggest that when production processes are divided into many stages operated by distinct and distant

firms, it becomes inherently harder for them to implement a sufficient level of risk-mitigating measures. Risks that were looked after by a few agents are transferred to the system, thus creating systemic risks and reducing economic resilience. Mitigating such risks can be seen as a common-good problem: as a whole, a supply chain clearly benefits from inventories, yet individual firms are not sufficiently incentivized to implement them.

The benefits of having more integrated decision-making processes along a supply chain have been recognized in the supply-chain management literature [41]. Coordination among firms could reincentivize risk-mitigating measures in fragmented supply chains. In practice, this could take various forms [42, 43]. For instance, bilateral supply-chain contracts, such as revenue-sharing agreements, could facilitate information sharing and directly extend the scope of risk management [44, 45]. Sales-rebate, buy-back, and quantity-flexibility contracts could lessen the financial risks of maintaining a safety stock [see 46 for a review]. Third-party orchestrators could help firms coordinate larger and more specialized subcomponents of their supply chains [47, 48].

Our work shows that, through precise supply-chain mapping, several simple indicators can be used to define the inventory levels that firms should keep to minimize systemic risks. In practice, which agents could use such benchmark levels? In financial systems, the central banks are legitimate decision-makers to deal with systemic risk, but no equivalent agents exist for supply chains. We suggest that three types of agents could be genuinely interested in such a top-down approach to systemic-risk mitigation. First, within large corporations, supply-chain managers could use this information to facilitate coordination between subsidiaries or production sites. Next, to secure the provision of critical goods, such as food or health-related products, policy makers could be interested in setting minimum inventory requirements within relevant supply chains. Last, insurers are increasingly asked to cover the economic losses generated by supply-chain disruptions. The use of indicators such as those proposed here could thus help them tailor their insurance policies to the position of firms in their supply chains.

Our results also highlight a trade-off between environmental performance and economic resilience. Supply chains can deliver a steadier flow of outputs in the face of external perturbations through the maintenance of inventories. This resilience comes at the cost of higher input consumption, leading to a loss of material efficiency. For instance, in our model, with the benchmark parameter values, the sales of an integrated supply chain can be 22% higher compared to a fragmented one while consuming 40% more raw materials, indicating a trade-off between a green [49] and a resilient supply chain [50]. This trade-off could be particularly marked for perishable goods, such as food. For instance, to ensure high on-shelf availability of food products [51], retailers might build larger stocks, inducing extra wastage [52, 53]. In a turbulent environment, securing a disruption-free supply of such goods can thus be an objective that conflicts with the mitigation of detrimental environmental impacts.

Last, our work demonstrates how evolutionary dynamics can help analyze agents' interactions on complex networks and their unfoldings. It focuses on supply risks and the ordering strategies of firms. Motivated by the surge in demand for specific products such as face masks in the COVID-19 pandemics, a future study could assess optimal inventory levels to cope with the demand uncertainty. This problem has been extensively addressed in operation research models [54, 55], but always in very simple, mostly two-layer, supply chain structures. The approach based on evolutionary dynamics on complex networks helps overcome this limitation and provide insights into more realistic supply chains.

## Supporting information

**S1 File.**
(DOCX)

## Acknowledgments

This study was enabled by the organizational support of the International Institute for Applied Systems Analysis (IIASA), through the Young Scientists Summer Program (YSSP).

## Author Contributions

**Conceptualization:** Célian Colon, Åke Brännström, Elena Rovenskaya, Ulf Dieckmann.

**Formal analysis:** Célian Colon, Åke Brännström.

**Investigation:** Célian Colon, Ulf Dieckmann.

**Methodology:** Célian Colon, Åke Brännström, Elena Rovenskaya, Ulf Dieckmann.

**Project administration:** Elena Rovenskaya, Ulf Dieckmann.

**Supervision:** Elena Rovenskaya, Ulf Dieckmann.

**Visualization:** Célian Colon.

**Writing – original draft:** Célian Colon.

**Writing – review & editing:** Célian Colon, Elena Rovenskaya, Ulf Dieckmann.

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
