## [Decision Letter · Decision Letter 0]

30 Jun 2020

PONE-D-20-14169

How fragmented supply chains amplify the impacts of extreme events and systemic risks

PLOS ONE

Dear Authors,

Thank you for submitting your manuscript to PLOS ONE. After careful consideration, we feel that it has merit but does not fully meet PLOS ONE’s publication criteria as it currently stands. Therefore, we invite you to submit a revised version of the manuscript that addresses the points raised during the review process.

 Please see comments below and additional attached comments of the reviewer.

We look forward to receiving your revised manuscript.

Kind regards,

Dejan Dragan, PhD

Academic Editor

PLOS ONE

Journal Requirements:

Additional Editor Comments:

The article was reviewed by three reviewers. Two of them require a major revision and the third a minor one. The unusual structure of the presentation of the sections of the paper is particularly emphasized. Therefore, it is suggested that the authors strictly follow the instructions and comments of the reviewers.

Reviewers' comments:

Reviewer's Responses to Questions

**Comments to the Author**

1. Is the manuscript technically sound, and do the data support the conclusions?

Reviewer #1: Yes

Reviewer #2: Yes

Reviewer #3: Partly

2. Has the statistical analysis been performed appropriately and rigorously? 

Reviewer #1: Yes

Reviewer #2: Yes

Reviewer #3: I Don't Know

3. Have the authors made all data underlying the findings in their manuscript fully available?

Reviewer #1: Yes

Reviewer #2: Yes

Reviewer #3: No

4. Is the manuscript presented in an intelligible fashion and written in standard English?

Reviewer #1: Yes

Reviewer #2: No

Reviewer #3: Yes

5. Review Comments to the Author

Reviewer #1: The authors propose a framework to analyze the effect of inventory size on the risk mitigation of unforeseen disruptions in supply chains. They model the flow of monetary value of material among the firms in a supply chain as an acyclic graph with the input node as raw material producing firms and the output as consumers. They analyze the effect of fragmentation, durability of goods, and productivity of the firms on the mitigation success achieved through optimal inventory decisions. The results are based on empirical average of simulated supply chains for various parameter combinations and graph configurations and provides valuable insight into the optimal inventory choices to mitigate the supply chain disruption risk. Simulation based analysis allows for analysis of more complex scenarios.

Overall the paper is well organized and the results are convincing. The methodology seems sounds and mathematically rigorous. I do have a couple of comments that I’d like to see addressed in a revised version of the manuscript.

How realistic are the parameter values considered for simulation? Are the number of firms and suppliers per firm realistic?

It’d be nice to add some mathematical definitions or explanations of some key terms (like mitigation success, productivity, failure rate) before discussing the results. It may be a good idea to move the model section before the results. The meaning of a few terms and the graphs used in the result section are only clear after reading the model section. I would suggest adding a table of mathematical symbols and their definitions.

The durability of input is assumed to be constant across the supply chain. It’s a simplifying assumption but I would be interested to know how could it affect the results if each firm has a fixed durability that could be an input to the simulation. Also, the demand has be assumed to be constant. Can this framework be used to analyze the effect of demand change due to disruptions? For instance, how could the industries have prepared for a sudden increase in demand for Lysol wipes or face masks given the current pandemic? It would be nice to have a framework to analyze the optimal inventory levels that could be prescribed for future demand changes in the event of a pandemic.

Simulations are expensive. Can one use variance reduction techniques (for instance importance sampling in Monte Carlo simulation for example) to reduce the computation time?

How well does the simulation scale with the number of suppliers in the chain? It would be nice to add the computation time, CPU usage, and memory requirements to run simulations.

Reviewer #2: The topic of the paper is current and the research itself is very interesting. However, the organisation of the paper is a bit unusual. Some results are mentioned imediatelly after the introduction part, this section is very brief and a lot of details are described in the suplementary materials section. This section if followed by the discussion, next chapter is a model with some more numerical experiments. I recommend to reorganize the whole paper into the more standard form to make it easier to read. Some parts of suplementary materials should be written directly in the paper to explain some interesting details about the research.

Reviewer #3: See the report in the attachment.

See the report in the attachment.

See the report in the attachment.

See the report in the attachment.

See the report in the attachment.

See the report in the attachment.

See the report in the attachment.

6. PLOS authors have the option to publish the peer review history of their article (what does this mean?). If published, this will include your full peer review and any attached files.

Reviewer #1: No

Reviewer #2: No

Reviewer #3: No

---

## [Author Response · Author response to Decision Letter 0]

30 Nov 2020

Reviewer #1: The authors propose a framework to analyze the effect of inventory size on the risk mitigation of unforeseen disruptions in supply chains. They model the flow of monetary value of material among the firms in a supply chain as an acyclic graph with the input node as raw material producing firms and the output as consumers. They analyze the effect of fragmentation, durability of goods, and productivity of the firms on the mitigation success achieved through optimal inventory decisions. The results are based on empirical average of simulated supply chains for various parameter combinations and graph configurations and provides valuable insight into the optimal inventory choices to mitigate the supply chain disruption risk. Simulation based analysis allows for analysis of more complex scenarios.

Reviewer #1: Overall the paper is well organized and the results are convincing. The methodology seems sounds and mathematically rigorous. I do have a couple of comments that I’d like to see addressed in a revised version of the manuscript.

Reviewer #1: How realistic are the parameter values considered for simulation? Are the number of firms and suppliers per firm realistic?

Response: The number of firms considered to be within a supply chain can significantly vary according to the definition of a supply chain and the type of final product. One approach considers that any service or good provided to a company is part of its supply chain—e.g., for a car rental company, it would include the electricity provider and the manufacturer of the computers used by the employees. Other approaches consider that only the suppliers providing significant parts of the final goods are in the supply chain—e.g., for a rice producer, only the farmer providing the paddy rice will be included, sometimes also the packaging supplier, but not the machine producer. The number of firms within the supply chain will be much higher in the first approach than in the second.

Our model uses the second approach, in which we only consider up to four of the most significant suppliers. Nation-wide datasets on firm-level supplier networks were extensively analyzed in Japan; see Fujiwara and Aoyama 2010 (reference in the text), Ohnishi, T., Takayasu, H., Takayasu, M., 2010. Network motifs in an inter-firm network. J Econ Interact Coord 5, 171–180, Mizuno, T., Souma, W., Watanabe, T., 2014. The structure and evolution of buyer-supplier networks. PLoS ONE 9, e100712. These data include the main suppliers of firms that were as crucial to credit rating companies. The average number of suppliers per firm is about 4. Another dataset is that of CompuStat for the USA, in which firms report the clients that represent more than 10% of their yearly sales. This dataset was analyzed by Barrot and Sauvagnat 2016 (reference in the text) and Atalay, E., Hortaçsu, A., Roberts, J., Syverson, C., 2011. Network structure of production. PNAS 108, 5199–5202. They found that firm had about 3.5 suppliers on average. These data concern Japan and the USA, which are the most advanced economies and in which the number of suppliers per firm is likely to be higher than the world average. Our choice of one to four suppliers per firm is, therefore, reasonable. We added these information in lines 191-193.

Concerning the length of the supply chains—which is less dependent on the definition of supply chains than the number of firms—our model uses between 3 and 6 layers, calculated using the trophic level approach mentioned in the manuscript. This number of layers, or tiers in the supply-chain literature, seems reasonable. Food supply chains would typically consist of three layers: farmers, processors, retailers (or more if you include wholesalers and cooperatives). Electronic supply chains could consist of five or more: mines, metal processors, electronic component manufacturers (1, 2, or 3 layers), assemblers, retailers. We added the information on the number of layers, lines 193-195.

The three firm-level parameters of our model are productivity, durability, and failure rate. Since we are interested in the qualitative behavior of the model, we explore the full space of these three parameters. We added these precision lines 196-199. 

We illustrate our results using benchmark values: z=2, v=50%, and p=10%. 

 Productivity z is defined as the ratio outputs over inputs. We analytically explored the full parameter space z>=1 in simple networks (SI Sections S1 to S4) and numerically explored 1>=z>=5. We use 2 as benchmark value. This value is consistent with the average output over input ratio found in the national input-output tables. For instance, using the 2015 input-output table for the USA provided by the OECD (https://stats.oecd.org/), the sector-average ratio of output over input is 2.2. It ranges between 1.3 and 4.2 across sectors.

 Durability is defined by the share of inventories that persists over a time step. We explore the full range of durability, from 0% to 100%. It is hard to come up with a comparable empirical metric since durability ranges from almost 0% for some food products and almost 100% for some raw metal products. That is why we systematically explored the full range and use 50% as a benchmark value.

 Failure rate is the probability that at each time step a firm becomes unable to produce. We explore the full range, from 0% to 100%. Failure has many causes: disasters, strike, workers’ health, administrative decisions. Our choice of 10% as benchmark value would mean that, on average, firms face such perturbation 5.2 weeks per year, which is a reasonable number.

We added a summary of this discussion, lines 191-195.

Reviewer #1: It’d be nice to add some mathematical definitions or explanations of some key terms (like mitigation success, productivity, failure rate) before discussing the results. It may be a good idea to move the model section before the results. The meaning of a few terms and the graphs used in the result section are only clear after reading the model section. I would suggest adding a table of mathematical symbols and their definitions.

Response: We moved the model section before the results. The definition of mitigation success is given in equation (2), that of productivity is given in lines 117-118 (we reworded it), that of failure rate at line 133-134. We added a table of mathematical symbols and their definitions (Table 1).

Reviewer #1: The durability of input is assumed to be constant across the supply chain. It’s a simplifying assumption but I would be interested to know how could it affect the results if each firm has a fixed durability that could be an input to the simulation.

Response: We agree that adding heterogeneity in firm-level parameters is an exciting suggestion. In reality, firms are heterogeneous and such heterogeneity has an impact on the aggregate behavior of the economy (see Gabaix, X., 2011. The granular origins of aggregate fluctuations. Econometrica 79, 733–772, or our contribution Colon & Ghil 2017, reference in the manuscript).

In the model, we have tested heterogeneous combinations of productivity, failure rate, and durability, as mentioned at the end of the model section, lines 197-199. These tests were used to check the robustness of the main findings, in particular the risk-increasing effect of fragmentation. We carried out firm-level analyses to analyze how their strategy change according to their position in the supply chains. Such analyzes underpin Figure 1c and Figure 5. 

It would be interesting to analyze how the strategy of specific firms change according to the durability of their inputs or that of their suppliers and clients. We did not perform this firm-level analysis, which we think is going out of the scope of the study as we have framed it. Some of our analytical work presented in the Supplementary Sections S1, S2, S3, and S4 does, however, provide some insights. We show that, for an individual firm, durability enables the range of conditions under which overordering is profitable. We expect that firms with more durable inputs will overorder more. We also expect that the more downstream those firms are—e.g., if they are final producers—the larger the benefits for the whole supply chain. In other words, durable final products matter more than durable raw materials.

Reviewer #1: Also, the demand has be assumed to be constant. Can this framework be used to analyze the effect of demand change due to disruptions? For instance, how could the industries have prepared for a sudden increase in demand for Lysol wipes or face masks given the current pandemic? It would be nice to have a framework to analyze the optimal inventory levels that could be prescribed for future demand changes in the event of a pandemic.

Response: There is a whole stream of operation research literature that deals explicitly with optimal ordering and inventory management in the face of demand uncertainty (see, for instance, the literature on the newsvendor problem, e.g., Kim, G., Wu, K., Huang, E., 2015. Optimal inventory control in a multi-period newsvendor problem with non-stationary demand. Advanced Engineering Informatics 29, 139–145, and the literature on the bullwhip effect, e.g., Riddalls, C.E., Bennett, S., 2002. The stability of supply chains. International Journal of Production Research 40, 459–475 and Lee, H.L., Padmanabhan, V., Whang, S., 1997. The bullwhip effect in supply chains. Sloan Management Review 38, 93–102).

To our knowledge, our approach, which is based on evolutionary dynamics on complex supply chain structures, has not been applied to this problem. It would be interesting to extend the model and include this aspect. We have decided not to do so at this stage. The model, as such already present a rich behavior, which we have tried to comprehensively analyze in this manuscript before moving to a more complex version. Such an endeavor would constitute an interesting follow-up, which we mention in the new paragraph at the end of the discussion section. 

Reviewer #1: Simulations are expensive. Can one use variance reduction techniques (for instance importance sampling in Monte Carlo simulation for example) to reduce the computation time?

Response: The variance reduction technique called common random numbers is already implemented in the evolutionary process. When a firm tries three overordering values and evaluate their impact on profit, the same sequence of perturbations is used in the three tests, rather than randomly generating three sequences of perturbations. We observed a reduction in variance indeed. We added the name of the variance reduction technique in Supplementary Section S6 where the process is explained, lines 244-245.

Importance sampling requires designing an appropriate biased distribution of the random input variables, which overemphasize the “important” region of the input space. In our cases, the random variables are the perturbations applied to the firms. It is not clear whether oversampling specific sequences of perturbations could accelerate the convergence of the evolutionary process or lead to lower variance estimates of the evolutionary stable strategies. The network structure makes it very unclear which sequence of perturbations have more impact than others. We would, of course, welcome any insights on this matter.

Reviewer #1: How well does the simulation scale with the number of suppliers in the chain? It would be nice to add the computation time, CPU usage, and memory requirements to run simulations.

Response: We rerun simulations for a fully fragmented supply chain with 30 firms and 2 suppliers per firm. It takes about 70 seconds to reach the evolutionary equilibrium on a laptop with an Intel i7-7500U CPU @ 2.70 GHz. Memory usage is about 68 Mo, and CPU usage reached 35%. The model is coded on MatLab language and run on Matab or Octave. The computation time scales superlinearly with n, the number of firms in the chain. It seems to scale at n*ln(n). We added this information to the Supplementary Information, lines 279-282.

Reviewer #2: The topic of the paper is current and the research itself is very interesting. However, the organisation of the paper is a bit unusual. Some results are mentioned imediatelly after the introduction part, this section is very brief and a lot of details are described in the suplementary materials section. This section if followed by the discussion, next chapter is a model with some more numerical experiments. I recommend to reorganize the whole paper into the more standard form to make it easier to read. Some parts of suplementary materials should be written directly in the paper to explain some interesting details about the research.

Response: We reorganized the whole paper into the more standard form: introduction, method/model, result, discussion.

We reviewed the Supplementary Information (SI) and investigated which sections could go into the main text. We believe that they are appropriately placed in the SI; we explain the reasons in the following paragraphs. We are, of course, open to reevaluating this if you have specific suggestions.

Sections S1 to S5 provide additional results obtained analytically. These results are useful to understand the behavior of the model better, however, they concern only extreme cases (e.g., fully perishable goods) in specific networks (e.g., one producer and its supplier). Because these results are not necessary to establish the main findings, we believe that they are appropriately placed in the SI. Since they consist of mathematical demonstrations and equations, we believe that, if placed in the result section of the main text, they could unnecessarily discourage potential less-technical audience from reading the paper.

Sections S6 and S7 provide details of the evolutionary process. They give useful details for colleagues specifically interested in evolutionary dynamics and algorithms. As such, they are not essential to understand the method and results. Since they are clearly flagged in the method section, we believe that colleagues that want to know more about the evolutionary process can easily access this extra information.

Section S8 shows a result obtained with the algorithm described in Section S5 for a layered network. The periodicity found in this case is an interesting example of unexpected patterns emerging from local interactions in complex systems. It recalls, among other things, the intriguing property of spin glasses. But, because it sets off in a particular class of network, this result does not add much to the main topic of the paper. That is why we put it in the SI. We mention this result at the beginning of the result section, such that mathematicians and physicists interested in complex systems can find it easily.

Section S9 shows the dispersion of the results shown in Figure 3. It is merely a robustness check and, as such, fits appropriately in the SI. Section S10 presents the details of the simple regression carried out between overordering rates and network indicators. The main results are given in the main text, whereas Section S10 provides methodological details and reports the full results.

Reviewer #3: See the report in the attachment.

The authors introduce a stylized model in which firms mitigate the risks of disruptions in their supply chains using inventories.

The subject of the paper fits with the aim of the PLOS ONE journal. The paper addresses a subject that has a large audience. The results are interesting and worthy of publication in PLOS ONE journal. 

Suggestions for paper improvement.

Reviewer #3: 1. I suggest the authors to give a number to each section. Why the Conclusion section is missing ?

Response: We follow a very classical structure: introduction, model, results, discussion. The conclusion section is integrated within the discussion section. Following the PLOS ONE guidelines (https://journals.plos.org/plosone/s/submission-guidelines), we have created “a mixed Discussion/Conclusions section (commonly labeled “Discussion”)”. 

We do not think that numbering the section would add much to the readability of the manuscript. We looked at the papers featured on the PLOS ONE homepage, and none of them had numbered sections. We would, of course welcome suggestions of the editors on this matter.

Reviewer #3: 2. I suggest a change of the title of the paper. See for example the following suggestions: An analysis of extreme events impact on systemic risks of fragmented supply chains. How extreme events impact the increase of the systemic risks of fragmented supply chains

Response: We changed the title of the paper to “Fragmentation of production amplifies systemic risks in from extreme events supply-chain networks”. It highlights the main finding of the result. 

Reviewer #3: 3. Abstract. 

Response: As suggested we completely rewrote the abstract. We respond to your specific comments below.

Reviewer #3: The authors write "Inventories greatly reduce disruption cascades and total costs at the expense of a moderate increase in direct losses." The proposition is incomprehensible. "Total costs" refers to what ? Inventories reduce the total costs ? The following is incomprehensible "at the expense of a moderate increase in direct losses."

Response: We replaced “total costs” with “indirect losses” in this sentence. We introduce in the second sentence of the abstract the concept of direct and indirect losses. We believe that it gives the necessary context for the readers to understand that sentence.

Reviewer #3: " Incentives to maintain inventories are progressively reduced in more fragmented supply chains." ???

Response: We rephrased it as follows: “The more fragmented a supply chain is, the less beneficial it is for individual firms that compose it to maintain inventories, resulting in higher systemic risks”.

Reviewer #3: "Risks are transferred from individual firms to the system. As a result, systemic risk builds up." I am asking the authors to give details what they understand by system. To what system they refer ?

Response: We removed the word “system” which may be confusing. The term “systemic risks” is introduced a few sentences above.

Reviewer #3: Please revise the last sentence of the abstract, that is " If insurers—or any other decision-makers concerned with that issue—were to tackle systemic risks by prescribing inventories to firms, estimating optimal sizes based on network analysis is substantially superior to any one-size-fits-all values." To which systems refers "systemic risk" ?

Response: We cut this sentence in two to increase readability. The term “systemic risk” is introduced a few sentences above: “systemic risks in supply chain networks”. It should therefore be clear to the reader that we are writing about systemic risks in supply chain networks.

Reviewer #3: 4. Pg 6. Line 210-211. The definitions of firm failure and durability are missing.

Response: We fixed this by interchanging the result and model sections. The failure rate is defined line 133-134. Durability is defined lines 120-122.

Reviewer #3: The authors write: "Supply chains can become unproductive when ." They do not give any reason for that.

Response: When input is 1, expected output is 0*p + z*(1-p). When p >= 1-1/z, the expected output is lower than the input. In that case, firms are unproductive. That is what we describe lines 325-329:

“First, when the failure rate p is high compared to the productivity z, supply chains can become unproductive. This occurs when p >= 1-1/z, independent of the durability v and the supply-chain structure; see the grey regions in the three panels of Fig. 4. There, firms use on average more than one unit of input to produce one unit of output, i.e., they are unproductive.”

Reviewer #3: 5. Pg 9. Line 347. You should write because the reader does not understand the definition of di.

Response: We made the suggested modification.

Reviewer #3: 6. Pg 9. Line 351-353. One time the authors say that are quantities and other time they say that are expressed in the same monetary terms. This is a contradiction.

Response: It is a common practice to quantify goods in monetary terms. For instance, we can say 600 USD worth of aluminum. Input-output tables (see for instance the World Input Output Database) and trade data (see for instance the UNComTrade data) express quantities of commodities in monetary terms. In our model, since we are not modeling prices, there is a formal equivalence between physical units and monetary units.

Reviewer #3: 7. Pg 10. Line 2. Please write "We consider the input–output matrix , where…"

Response: We made the suggested modification. We also make the same change for the adjacency matrix M.

Reviewer #3: 8. Pg 10. Line 3. Please explain why in the following equation occurs z*si

Response: It is because firms equally divide the total amount of inputs they need between their suppliers. We rephrased lines 125-126 to make this clearer.

Reviewer #3: 8. Pg 10. Line 5. Please define 

Response: We made the suggested modification.

Reviewer #3: 9. Pg 10. Explain how are computed and .

Response: Profit is defined as sales minus input costs at lines 128 and 129. At these lines, we show the formula for \\{pi}_{t,0}, which corresponds to the case without overordering and without perturbation. We simplified the formula so that readers can visually identify sales, y_i, and input costs, y_i/(z*s_i). We clarified the definition of the profit at each time step \\{pi}_{t,i} at lines 148-149. We added the formula which defines the average profit over a time horizon T from \\{pi}_{t,i} at line 151.

Reviewer #3: The reviewer does not understand the order of sections of the paper. First the authors present the results and discussions and then the model is described. This is not normal. I think that the first section should be the introduction. In section 2 should be described the model. Then the results and analysis of the results should be presented.

Response: We made the suggested modification.

Reviewer #3: In the paper is not clear what are the input data of the model and which is their origin. I am asking the authors to explain how the results analysis follows from their model.

Response: This model is a theoretical exercise, often called “stylized model” or “toy model”, which aims to qualitatively assess the interactions between system characteristics, here between supply-chain fragmentation and disruption-risk reduction. Such an approach is often used to form hypotheses on the behavior of complex systems; see, for instance May, R.M., Levin, S.A., Sugihara, G., 2008. Complex systems: Ecology for bankers. Nature 451, 893–895. We do not directly use empirical data, nor are we aiming to reproduce observed time series quantitatively. We added a sentence at the end of the introduction section to highlight this approach.

The model is explained in the model section. The last subsection of this section, “Numerical and analytical investigations”, explains how the results are generated from the model. Numerically, we generate networks and systematically explore a subset of the 3-dimensional parameter space. Analytically, in other words, “with pen and paper”, we mathematically explore simpler network configurations. The result section presents the main findings of our numerical and analytical studies. The SI focuses on specific results and presents in detail the analytical work.

Reviewer #3: Taking into account the above comments I recommend the authors to make a major revision of their paper.

---

## [Editor Report · Decision Letter 1]

7 Dec 2020

Fragmentation of production amplifies systemic risks from extreme events in supply-chain networks

PONE-D-20-14169R1

Dear Authors,

We’re pleased to inform you that your manuscript has been judged scientifically suitable for publication and will be formally accepted for publication once it meets all outstanding technical requirements.

Kind regards,

Dejan Dragan, PhD

Academic Editor

PLOS ONE

Additional Editor Comments (optional):

All comments were appropriately followed in the paper. Accordingly, the paper deserves an opportunity to be accepted. AE DD
---

## [Editor Report · Acceptance letter]

16 Dec 2020

PONE-D-20-14169R1 

Fragmentation of production amplifies systemic risks from extreme events in supply-chain networks 

Dear Dr. Colon:

I'm pleased to inform you that your manuscript has been deemed suitable for publication in PLOS ONE. Congratulations! Your manuscript is now with our production department. 

Kind regards, 

on behalf of

Dr. Dejan Dragan 

Academic Editor

PLOS ONE